# Evaluation of Soil-Applied Chemical Fungicide and Biofungicide for Control of the *Fusarium* Wilt of Chrysanthemum and Their Effects on Rhizosphere Soil Microbiota

**Huijie Chen [1,2], Shuang Zhao [1,2], Kaikai Zhang [1,2], Jiamiao Zhao [1,2], Jing Jiang [1,2], Fadi Chen [1,2] and Weimin Fang [1,2],***

[1]   College of Horticulture, Nanjing Agricultural University, Nanjing 210095, China;
      2016204031@njau.edu.cn (H.C.); zhaoshuang@njau.edu.cn (S.Z.); 2014104105@njau.edu.cn (K.Z.);
      2017104106@njau.edu.cn (J.Z.); 2017804128@njau.edu.cn (J.J.); chenfadinjau@163.com (F.C.)
[2]   Key Laboratory of Landscaping, Ministry of Agriculture, Nanjing 210095, China
*    Correspondence: fangwm@njau.edu.cn; Tel.: +86-025-8439-5231

**Abstract:** Chemical fungicides are a frequently used intervention for the control of the *Fusarium* wilt of chrysanthemum, but are no longer considered environmentally friendly. However, the biofungicides offer one of the best alternatives to reduce the use of chemical fungicides. In this study, a series of two-year greenhouse experiments were conducted to evaluate the soil-applied chemical fungicide (dazomet, DZ) and biofungicide (biocontrol agent combined with *B. subtilis* NCD-2, BF) for controlling the *Fusarium* wilt of chrysanthemum and its effects on rhizosphere soil microbiota. The results indicated that DZ and BF showed good control efficacy of *Fusarium* wilt of chrysanthemum in the two-year application evaluation. However, the DZ treatment significantly decreased the soil catalase and urease activities compared with the control, while BF showed a significant increase in bacterium/fungus ratios (B/F), soil urease and acid phosphatase activities. Abundances of potential plant pathogens *F. oxysporum*, *Rhizoctonia zeae* and *Rhizoctonia solani* were also lower, while potential plant-growth-promoting micro-organisms like the *Rhizobiales* bacterium and *Mariniflexile* sp. were higher in the BF treatment than in the control. Our findings suggested that the overall effect of the soil biota on chrysanthemum growth was more positive and stronger in the BF treatment than in the DZ treatment.

**Keywords:** *Bacillus subtilis*; *Fusarium* wilt; cut chrysanthemum; dazomet; soil microbe; soil enzymatic activities

## 1. Introduction

Chrysanthemum (*Chrysanthemum morifolium*, Asteraceae, Plantae) is an important and popular ornamental plant all over the world [1]. The demand for chrysanthemum increases by improved standards of living, prompting a rise in the monoculture-based production of chrysanthemum [2]. However, production of chrysanthemum is severely hampered by *Fusarium* wilt, a soilborne wilt infected by *Fusarium oxysporum* f. sp. *chrysanthemi* (*Fusarium oxysporum*), which causes significant decrease in the yield and quality of chrysanthemum [2].

*Fusarium oxysporum* (Hypocreales, Ascomycota) is the causal agent of *Fusarium* wilt [3]. It has been reported to be one of the most destructive soilborne pathogens [4]. *Fusarium oxysporum* is widely distributed in various soil types worldwide [5]. This pathogen can grow and survive in soil for 10–15 years and infect more than one hundred plants, including the cucumber (*Cucumis sativus* L.) [5],

watermelon (*Citrullus* L.) [6,7], cotton (*Gossypium* spp.) [8], and banana (*Musa* nana L.) [9,10]. Susceptible plants can be infected through the root during all stages of growth, causing disease of the vascular bundle, gradually turning leaves yellow, and eventually resulting in plant wilting [11].

In facility cultivation, *Fusarium* wilt is usually controlled by commercially available chemical fungicides such as methyl bromide [12], carbendazim [13], and dazomet [14]. However, chemical fungicides are considered environmentally unfriendly, and their sustained use can result in soil ecosystem problems [15]. The biological control of *Fusarium* wilt has advanced greatly over the past decade. Controlling *Fusarium* wilt by antagonistic microorganisms represents an alternative disease management strategy due their ability to provide environmentally friendly disease-control efficacy [16]. The application of bioeffector agents such as *Bacillus subtilis* and *Paenibacillus polymyxa* (both Firmicutes, Bacteria) and *Trichoderma harzianum* (Hypocreales, Ascomycota) has been widely used to suppress soilborne pathogens [17–20]. Factors contributing to their disease-control properties include their ability to produce various antimicrobial compounds [21], their efficient colonization of the root surface for long periods of time [22], and their long-term viability, which makes it feasible to develop commercial products [23].

It is crucial to maintain soil microbial community and diversity for soil health [24]. Soil-enzyme activities are often used as sensors in studies on the influence of soil treatments on microbial activity and soil fertility [25]. And soil enzyme activities have been reported to be greatly affected by soil microbial properties because transformations of the important organic elements occur through microorganisms [26]. It is reported that the biofungicide had no significant impact on the wine microbiota, whereas the chemical fungicide caused a reduction of microbial community richness and diversity [27].

Most previous studies on the effects of chemical fungicides and biofungicides have focused on the prevention rate of *Fusarium* wilt and on yield and yield-related parameters [28], but very few are known about the effect of chemical fungicide or biofungicide application on soil enzymatic activity and diversity of rhizosphere soil microbiota. The large-scale use of chemical fungicides on ornamental plants has resulted in a series of soil problems, including soil pollution, biodiversity decrease, and pathogen resistance [19]. Thus, in the present study, the effects of soil-applied chemical fungicide and biofungicide on *Fusarium* wilt and the soil-enzyme activities and microbe of chrysanthemum were investigated. The results of this study can provide guidance for the control of *Fusarium* wilt of chrysanthemum, improve the knowledge of the composition of microbe communities in rhizosphere soil after chemical fungicide and biofungicide applications, and lead to a better understanding of microbe roles in these soil ecosystems.

## 2. Materials and Methods

### 2.1. Site Description and Plant Material

This investigation was conducted twice, from May to October in both 2015 and 2016, at the Nanjing Agricultural University "Chinese chrysanthemum germplasm resources conservation center", Nanjing, China, where chrysanthemums with a five-year history of continuous monocropping suffered from severe *Fusarium* wilt. Prior to the initiation of the experiment, the soil was sandy loam, had a pH of 6.96, and a specific conductance of 467.67 $\mu S\cdot cm^{-1}$, and contained 11.60 g organic matter·$kg^{-1}$, 0.09 g available N·$kg^{-1}$, 0.36 g available K·$kg^{-1}$, and 0.18 g available P·$kg^{-1}$. Seedlings of the chrysanthemum cultivar '*Jimba*' (provided by Honghua Horticulture Co. Ltd., Shanghai, China) were established in a greenhouse in a perlite medium at a spacing of 20 cm for 3 weeks under a 16 h photoperiod, with daytime temperature held at 28 °C and night-time temperature at 22 °C.

### 2.2. Fungicide Preparation and Preassessment of the Inhibition of F. Oxysporum Growth

Dazomet (3,5-dimethyl-1,3,5-thiadiazinane-2-thione, DZ, pure $\geq$ 95.0%) was bought from the Nantong Shizhuang Chemical Co., Ltd., Nantong, Jiangsu, China. DZ is a granular fumigant that releases methyl isothiocyanate, which is often used to treat soil before chrysanthemum replanting.

The biofungicide 'Xin Zhi Nong' ($10^9$ spores·g$^{-1}$ *Bacillus subtilis* NCD-2 wettable powder, BF) was purchased from the Baoding Kelvfeng Biochemical Technology Co., Ltd. (Baoding, Hebei, China). NCD-2 is highly efficient against multiple-plant pathogenic fungi (Guo et al., 2010; Guo et al., 2014). The strain *B. subtilis* NCD-2 was grown on liquid King's medium-B for 24 h as a shake culture at 150 rpm at room temperature (24 $\pm$ 1 °C). Bacterial suspensions were centrifuged for 20 min at 6520 g. Pellets were resuspended in 0.1 M MgSO4 in a ratio of 1:1 (*w/v*). Bacterial suspensions in 0.1 m MgSO$_4$ were mixed with 10% (*v/v*) glycerol, which was used as a bacterial preservative. This suspension was then mixed with an equal volume of autoclaved 1.5% Na-alginate. A wetting agent, Ca-lignosulfonate, was added (7%, *w/w*) to the mixture. The resulting mixture was thinly spread over a glass sheet and air-dried in a laminar air-flow cabinet at 24 °C for 1 h to form a slightly moistened powder (15% moisture content).

Potato dextrose agar (PDA) medium contained 2% DZ or 2% BF. Then, 0.6-cm diameter agar blocks were cut from the precultured *F. oxysporum* isolate CFD-B2 (The pathogen *F. oxysporum* CFD-B2 was isolated from the rhizosphere soil of chrysanthemum '*Jimba*' with *Fusarium* wilt and was kept in the chrysanthemum laboratory of Nanjing Agricultural University), and the hyphae was seeded down to the center of the PDA plate containing the fungicide. Each fungicide treatment consisted of three replicate plates. The growth of *F. oxysporum* CFD-B2 on PDA plates was observed after incubation at 28 °C for 4 days. Then, cuts were made in the roots of the '*Jimba*' seedlings by scissors, and the *F. oxysporum* CFD-B2 (1 $\times$ 10$^5$ per gram of soil) was added by root irrigation according to Koch's rule.

### 2.3. Field Experimental Design

The greenhouse experiments were performed twice from May to October of both 2015 and 2016. There were three treatments, each with three replicate plots: (1) an untreated control (Control); (2) 30 g dazomet·m$^{-2}$ soil (DZ); (3) 30 g biofungicide·m$^{-2}$ soil (BF). Each plot was 80 $\times$ 40 cm, holding 24 plants. The soil was plowed to a 20 cm depth before planting.

### 2.4. Experimental Methods

#### 2.4.1. Measurement of Chrysanthemum Growth and Quality

Eight plants were randomly sampled from each replicate plot 90 days after transplanting (October 2015 and 2016). Shoot height and diameter, shoot dry weight, root dry weight, leaf width and length, flower diameter, ray floret number, chlorophyll content (soil and plant analyzer 125 development—SPAD value), and nitrogen content in leaves were measured. Chlorophyll content and nitrogen content in leaves were measured using an SPAD-502 Plus chlorophyll meter (Top Instrument Co., Hangzhou, China) and a portable plant rapid nutrient-measuring instrument (TYS-3N, Top Instrument Co., Hangzhou, China).

#### 2.4.2. Disease Incidence

The symptoms of *Fusarium* wilt of chrysanthemum were checked for once a day. The disease incidence (DI) for each plot was assigned as the fraction of infected plants present at 90 days after transplanting, and the disease reduction percentage (DRP) was calculated by the formula $(1 - DT/DC) \times 100$. DC and DT represent the DI values in the control and treatment plots, respectively [7].

### 2.4.3. Soil-Enzyme Activity Analysis

Soil-catalase activity was measured using a titrimetric method [29]. Urease activity was measured using the sodium phenolate sodium hypochlorite colorimetric method [30]. Soil acid phosphatase activity was determined using a soil acid phosphatase activity Assay Kit (Suzhou Keming Biotechnology Co., Ltd., Suzhou, China) according to the manufacturer's protocol.

### 2.4.4. Soil DNA Extraction

The plant roots were carefully shaken, and the trace soil attached to the root was collected 90 days after transplanting (October 2015 and 2016); then, the soil samples were sieved through a 2 mm mesh and stored at $-80$ °C. Total DNA was extracted from the soil samples by using a Power Soil™ DNA Isolation kit (MoBio Laboratories, Carlsbad, CA, USA), and the extracted DNA was quantified using a NanoVue device (GE Life Sciences, Piscataway, NJ, USA).

### 2.4.5. Real-Time Polymerase Chain Reaction (PCR) Amplification

The abundance of bacterial and fungal species was quantified by an iCycler thermocycler with an iQ5 multicolor real-time PCR detection system (Version 2.1, Bio-Rad, Hercules, CA, USA). Amplification was performed in 20 μL reaction mixtures using SYBR Premix ExTaq II following manufacturer's instructions (Takara Bio, Otsu, Shiga, Japan). Two microliters of the original DNA extract were added as the template in each reaction mixture. The primer sets Eub338/Eub518 [31] (for bacteria), ITS1F/ITS4 [32] (for fungi) and ITS1-F/AFP308R [33] (for *F. oxysporum*) were used for quantification. The PCR regime comprised a 9 °C/5 min denaturation, followed by 40 cycles of 95 °C/15 s, 55 °C/30 s, and 72 °C/45 s, and a final extension of 72 °C/10 min. Following the thermal profile, melting curve analysis was performed to confirm the specificity of the PCR product in each real-time PCR amplification by continuously measuring fluorescence as the temperature increased from 55 to 95 °C. The parameter Ct (threshold cycle) was determined as the cycle number at which the start of an exponential increase in the reporter fluorescence was detected.

### 2.4.6. DGGE Profiling

The 18S rRNA and 16S rRNA amplification were performed in 25 μL reaction mixtures with the original DNA extracts as the template in each reaction mixture. The 25 μL reaction mixtures contained $1 \times$ ExTaq PCR buffer ($MgCl_2$), 100 mM dNTP, 0.5 μM each primer, 1U ExTaq DNA polymerase (Takara), 50 ng of DNA template and the primer set ITS1-GC/ITS4 (fungi) or Eub338-GC/Eub518 (bacteria). A GC clamp (5′-CGCCCGCCGCGCGCGGCGGGCGGGGCGGGGG ACGGGGGG) was added to the 5′ end of both ITS1F and Eub338 to stabilize the melting behavior of the amplicons. The PCR conditions consisted of a 94 °C/5 min denaturation, followed by 30 cycles of 94 °C/30 s, 56 °C/30 s, and 72 °C/60 s, and completed by a final extension of 72 °C /10 min. PCR products were electrophoresed on a 1.2% agarose gel and visualized under UV light by ethidium bromide staining.

A 20 μL aliquot of the PCR product (containing 400 ng of DNA) was loaded onto each lane of a denaturing gradient gel electrophoresis (DGGE) gel. For the fungal amplicons, the polyacrylamide concentration was 8% (37.5:1 acrylamide/bisacrylamide), and the denaturing gradient was 15%–40%. For the bacterial amplicons, the polyacrylamide concentration was 8% (37.5:1 acrylamide/bisacrylamide), and the denaturing gradient was 40%–60%. The electrolyte was 40 mM Tris acetate, 1 mM EDTA, pH 8.0. The temperature was held at 60 °C, and the voltage was 120 V for 10 min and then 80 V for 16 h. The amplicons were visualized by silver staining. Prominent fragments were excised and sequenced.

### 2.5. Statistical Analyses

The data were analyzed by Excel 2007 (Microsoft, Redmond, WA, USA), and one-way analyses of variance (ANOVA) and a differential significance test (Duncan multiple range test, $p < 0.05$) were

performed on SPSS 20.0 software (SPSS, Chicago, IL, USA). Silver-stained DGGE bands were analyzed with Quantity One computer software (version 4.6.3, Bio-Rad, Hercules, CA, USA). Each stripe pattern represented a single operational taxonomic unit, the intensity of each band was detected, and the system was analyzed according to an unweighted pair-group method with 192 arithmetic means (UPGMA) clustering analysis algorithm. Richness (S) is the number of different bands in the DGGE profiles. A Shannon index ($H'$) was calculated by $-\sum(P_i)\cdot(\ln P_i)$, $P_i = n_i/N$, where $n_i$ represents the peak intensity for a band. N is the sum of all peak intensities in the lane where the band is located. Index J was derived from the expression $1/\sum (P_i)^2$.

## 3. Results

### 3.1. Fungicide Preassessment

The growth of *F. oxysporum* CFD-B2 was observed on PDA plates with no fungicide or with 2% fungicide, and the inoculation colony of *F. oxysporum* CFD-B2 was 0.6 cm in diameter. During 24 days of culture, the growth of *F. oxysporum* CFD-B2 was greatly inhibited by DZ and BF; the diameter of the *F. oxysporum* CFD-B2 colony grown on PDA (control) was 7.33 cm, while its growth was 2.30 and 0.80 cm on the PDA plate with 2% BF and 2% DZ, respectively. The inhibitory effects of DZ were greater than those of BF (Figure 1a). The pot preassessment result indicated that, after the '*Jimba*' seedling substrate was inoculated with the *F. oxysporum* CFD-B2 spores, the leaves of the '*Jimba*' seedlings turned yellow from the bottom to the top of plants in the control treatment within 21 days, while the '*Jimba*' seedlings had no obvious symptoms in the both DZ and BF treatments (Figure 1b).

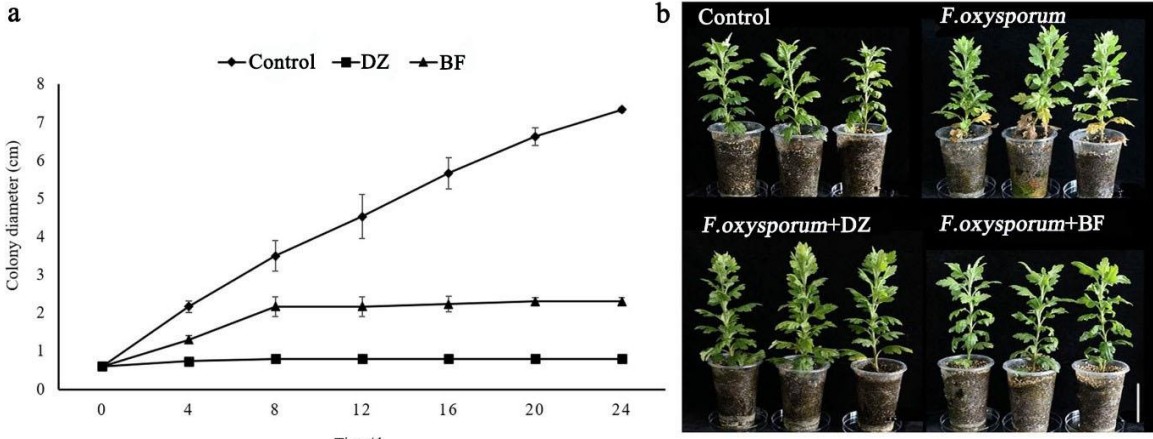

**Figure 1.** Effects of fungicides on the growth of *F. oxysporum* CFD-B2. (**a**) Colony diameter of *F. oxysporum* CFD-B2 on the Potato dextrose agar (PDA) plate with biocontrol agent combined with *B. subtilis* NCD-2 (BF) and dazomet (DZ) 24 days after inoculation; (**b**) Growth of chrysanthemum 21 days after planting; bar = 5 cm. Note: Control, the PDA, and seedling substrates had no added fungicide; DZ, the PDA, and seedling substrates had 2% dazomet; BF and the PDA had 2% biofungicide.

### 3.2. Effects of Fungicides on Chrysanthemum Fusarium Wilt

The DZ and BF significantly increased the disease-reduction percentage (DRP) and decreased the incidence of *Fusarium* wilt (DI) of chrysanthemum '*Jimba*' in the two-year evaluation (Figure 2). After the 90 day growth of the first cropping period (October 2015), the DI in the DZ treatment was 5.56%, while it was 8.33% in the BF treatment when compared with the control. After the 90 day growth of the second cropping period (October 2016), the DI was 15.28% and 6.94% in the DZ and BF treatments, respectively. The DRP of the DZ treatment was 63.04% in 2015 and 26.67% in 2016, while the DRP of the BF treatment was 44.57% in 2015 and 66.67% in 2016. The disease-control effect of BF in 2016 was greater than that in 2015, while the control effect of DZ in 2016 was less than that in 2015.

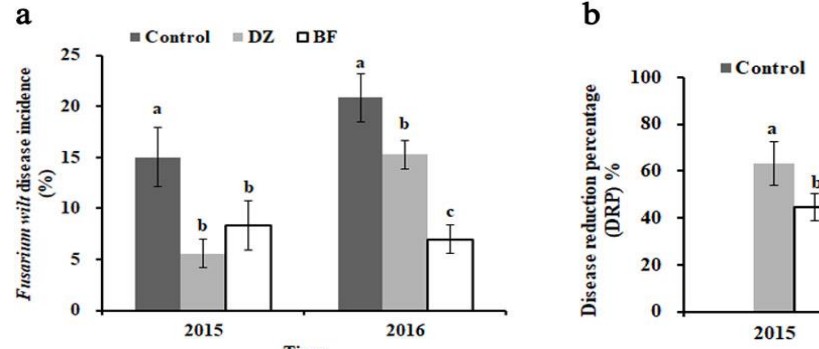

**Figure 2.** *Fusarium* wilt (**a**) incidence and (**b**) disease-reduction percentage of chrysanthemum '*Jimba*' treated with DZ and BF. Note: Bars represent the mean, and whiskers represent the standard deviation. Control: no treatment, DZ: 30 g dazomet per m² soil, BF: 30 g biofungicide per m² soil. Letters above the bars indicate a significant difference according to Duncan's multiple range test at the *p* < 0.05 level.

### 3.3. Effects of Fungicides on the Quality of Chrysanthemum

In 2015, the shoot height (71.25 cm) and shoot dry weight (7.20 g) were greatest in plants exposed to the DZ treatment; the shoot height increased 39.93% and 9.41% in the DZ and BF treatments compared with the control, respectively. In addition, leaf width (2.88 cm), flower diameter (12.82 cm) and flower dry weight (7.20 g) were greatest in the DZ treatment, followed by the BF treatment, while their lowest values occurring in the control (Table 1). Flower diameter increased 28.07% and 18.78% in the DZ and BF treatments, respectively, when compared with the control. There were no significant differences in shoot diameter, leaf SPAD value, nitrogen content, or root weight index. In 2016, shoot dry weight (4.63 g), root dry weight (0.29 g), flower diameter (12.36 cm), and flower dry weight (4.63 g) were greatest in the BF treatment, followed by the DZ treatment, and the control showed the lowest values. Compared with the control, the BF treatment showed a significant increase in the growth of shoots, leaves, roots, and flowers. Root dry weight and flower diameter increased 93.33% and 23.23%, respectively, in the BF treatment compared with the control in 2016, while, when compared with the control, the DZ treatment had no significant differences in leaf and flower index.

**Table 1.** Effects of chemical fungicide and biological fungicide on the growth of chrysanthemum.

| Growth Index | | 2015 | | | 2016 | | |
|---|---|---|---|---|---|---|---|
| | | Control | DZ | BF | Control | DZ | BF |
| Shoot | Height (cm) | 50.92 ± 2.39 [c] | 71.25 ± 1.54 [a] | 55.71 ± 1.15 [b] | 58.63 ± 4.92 [a] | 48.82 ± 3.73 [b] | 61.50 ± 2.18 [a] |
| | Diameter (cm) | 4.04 ± 0.17 [a] | 4.33 ± 0.31 [a] | 4.16 ± 0.03 [a] | 4.00 ± 0.26 [a] | 3.82 ± 0.24 [b] | 4.09 ± 0.07 [a] |
| | Dry wt (g) | 3.56 ± 0.26 [b] | 7.20 ± 0.89 [a] | 4.60 ± 0.21 [b] | 4.39 ± 0.07 [b] | 4.29 ± 0.07 [b] | 4.63 ± 0.06 [a] |
| Leaf | Width (cm) | 2.53 ± 0.10 [b] | 2.88 ± 0.14 [a] | 2.65 ± 0.03 [b] | 2.10 ± 0.07 [a] | 2.23 ± 0.06 [a] | 2.24 ± 0.14 [a] |
| | Length (cm) | 4.22 ± 0.04 [ab] | 4.48 ± 0.09 [b] | 4.57 ± 0.25 [a] | 3.50 ± 0.20 [a] | 3.53 ± 0.34 [a] | 3.78 ± 0.24 [a] |
| | SPAD Value (%) | 17.24 ± 0.42 [b] | 21.60 ± 0.32 [a] | 21.25 ± 0.29 [a] | 23.02 ± 0.21 [b] | 22.74 ± 0.12 [b] | 23.58 ± 0.24 [a] |
| | Nitrogen contents (mg/g) | 1.16 ± 0.02 [b] | 1.47 ± 0.02 [a] | 1.45 ± 0.02 [a] | 1.58 ± 0.02 [ab] | 1.56 ± 0.01 [b] | 1.62 ± 0.01 [a] |
| Root | Dry wt (g) | 0.13 ± 0.05 [a] | 0.22 ± 0.13 [a] | 0.25 ± 0.04 [a] | 0.15 ± 0.03 [c] | 0.23 ± 0.01 [b] | 0.29 ± 0.03 [a] |
| Flower | Flower diameter (cm) | 10.01 ± 0.02 [c] | 12.82 ± 0.20 [a] | 11.89 ± 0.08 [b] | 10.03 ± 0.19 [b] | 9.93 ± 0.26 [b] | 12.36 ± 0.46 [a] |
| | Dry wt (g) | 3.56 ± 0.26 [b] | 7.20 ± 0.89 [a] | 4.60 ± 0.21 [b] | 4.39 ± 0.07 [b] | 4.29 ± 0.07 [b] | 4.63 ± 0.06 [a] |

Data given in the form mean ± standard deviation. Treatment codes: Control: No treatment, DZ: 30 g dazomet per m² soil, BF: 30 g *Bacillus subtilis*-enhanced BF per m² soil. Different letters indicate significant differences among soil treatments according to Duncan's multiple range test at *p* < 0.05 level, [a], [b] and [c] indicated different levels of significant difference. SPAD—soil and plant analyzer 125 development

### 3.4. Effects of Fungicides on Chrysanthemum Soil Enzymatic Activities

Soil catalase, urease, and acid phosphatase activities in the rhizosphere soil of both treatments were significantly affected by fungicide application ($p < 0.05$) (Figure 3). Soil catalase activities decreased 42.56% in 2015 and 53.33% in 2016 in the DZ treatment when compared with the control. Soil catalase activities were not significantly different between the BF and control treatments in both 2015 and 2016 (Figure 3a).

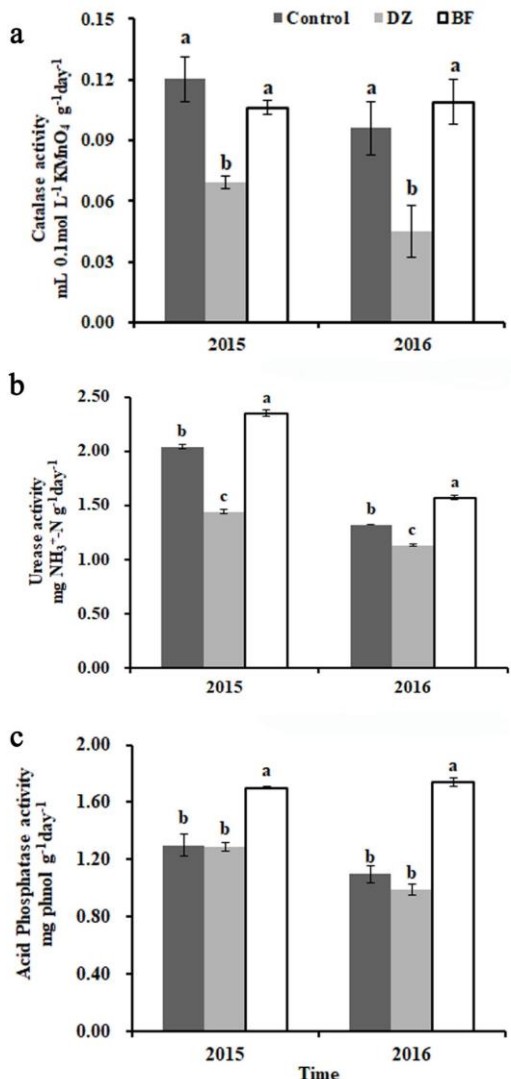

**Figure 3.** Enzyme activity. (**a**) Catalase; (**b**) urease; (**c**) acid phosphatase activity in rhizosphere soils of different treatment. Bars represent the mean, and whiskers represent the standard deviation.

Soil urease activities were 1.44 mg $NH_3{}^+$-N·$g^{-1}$·$day^{-1}$ in 2015 and 1.13 mg $NH_3{}^+$-N·$g^{-1}$·$day^{-1}$ in 2016 in the DZ treatments, which decreased 29.41% in 2015 and 14.39% in 2016 in the DZ treatments compared with the control. The urease activities increased 15.20% (in 2015) and 18.94% (in 2016), respectively, in the BF treatments compared with the control (Figure 3b).

Soil acid phosphate activities were increased by 30.77% in 2015 and 58.18% in 2016 in the BF treatments when compared with the control. There were no significant differences in this activity between the DZ and control treatments in both 2015 and 2016 (Figure 3c).

### 3.5. Effects of Fungicides on Soil Microbiota Detected by Real-Time PCR

The real-time quantitative results indicated that there was a significant difference ($p > 0.05$) in the colony-forming unit (CFU) levels of *F. oxysporum*, soil bacteria, and fungi in the rhizosphere soil of the DZ and BF treatments. In 2015, the abundance of *F. oxysporum* was 5.69 CFU·$g^{-1}$ rhizosphere soil in the control, 2.49 CFU·$g^{-1}$ rhizosphere soil in the BF-treated soil samples and 1.49 CFU·$g^{-1}$ rhizosphere soil in the DZ-treated soil samples 90 days after transplanting (Table 2). The abundance of *F. oxysporum* decreased by 56.24% and 73.81% in the BF- and DZ-treated soil samples in 2015 compared with the control. The abundance of *F. oxysporum* in DZ-treated soil samples in 2016 was not significantly different from those in the control. However, the abundance of *F. oxysporum* in BF-treated soil samples was $1.69 \times 10^3$ CFU·$g^{-1}$ soil, which showed a significant decrease in 2016 from 2015.

**Table 2.** Quantification of *F. oxysporum*, bacteria, and fungi in different treatments using real-time PCR.

| Time | Treatment | *F. oxysporum* ($10^3$ cfu·$g^{-1}$ soil) | Bacteria ($10^6$ cfu·$g^{-1}$ soil) | Fungi ($10^5$ cfu·$g^{-1}$ soil) | Bacteria/Fungi Ratio |
|------|-----------|------|------|------|------|
| 2015 | Control | 5.69 ± 0.09 [a] | 2.66 ± 0.16 [a] | 12.92 ± 0.25 [a] | 2.06 ± 0.59 [b] |
|      | DZ | 1.49 ± 0.21 [c] | 2.03 ± 0.18 [b] | 2.23 ± 0.61 [b] | 9.10 ± 0.93 [a] |
|      | BF | 2.49 ± 0.02 [b] | 2.15 ± 0.12 [b] | 2.96 ± 0.20 [b] | 7.26 ± 0.76 [a] |
| 2016 | Control | 6.61 ± 0.62 [a] | 2.50 ± 0.21 [a] | 15.10 ± 0.06 [a] | 1.65 ± 0.36 [b] |
|      | DZ | 6.03 ± 0.06 [a] | 1.64 ± 0.09 [b] | 0.61 ± 0.16 [b] | 1.51 ± 0.21 [b] |
|      | BF | 1.69 ± 0.07 [b] | 2.56 ± 0.16 [a] | 0.20 ± 0.54 [c] | 4.42 ± 0.76 [a] |

Data given in the form mean ± standard deviation. Treatment codes: Control: No treatment, DZ: 30 g dazomet per $m^2$ soil, BF: 30 g *Bacillus subtilis*-enhanced BF per $m^2$ soil. Different letters indicate significant differences among soil treatments according to Duncan's multiple range test at $p < 0.05$ level, [a], [b] and [c] indicated different levels of significant difference.

Based on real-time PCR outputs, both DZ and BF treatments led to a decreased abundance of bacteria and fungi, while cause a marked increase in the ratio of bacteria to fungi: up to 9.10 in the DZ treatment and 7.26 in the BF treatment in 2015 (Table 2). In 2016, the BF treatment was the most suppressive for fungi, also resulting in a significantly increase in the ratio of bacteria to fungi.

### 3.6. DGGE Fingerprints of DZ- and BF-Treated Rhizosphere Soil 16s and 18s Fragment

Based on DGGE profiles, both DZ and BF application significantly changed the community and diversity of soil microbes in 2015 and 2016. Both the 16s rRNA and 18s rRNA gene-amplified products from rhizosphere soil DNA gave distinct variations after DGGE profiling (Figure 4). Bands that were clearly separated from neighboring ones were easily visualized and isolated for subsequent sequencing. Considering these bands, the bacterial and fungal DGGE profiles fell into three clusters (Figure 4b,d,f,h). In 2015, the bacterial communities of BF clustered together with a similarity coefficient of 72% (Figure 4b), while they clustered together with a similarity coefficient of 74% (Figure 4d) in 2016. However, the fungal communities of BF clustered together with a similarity coefficient of 83% (Figure 4f) in 2015 and 52% in 2016 (Figure 4h). In addition, bacterial communities clustered with only 51% similarity between the DZ and BF treatments in 2015 (Figure 4b) and 58% similarity in 2016 (Figure 4d), whereas fungal communities only had 48% similarity between the DZ and BF treatments in 2015 (Figure 4f) and 34% similarity in 2016 (Figure 4h).

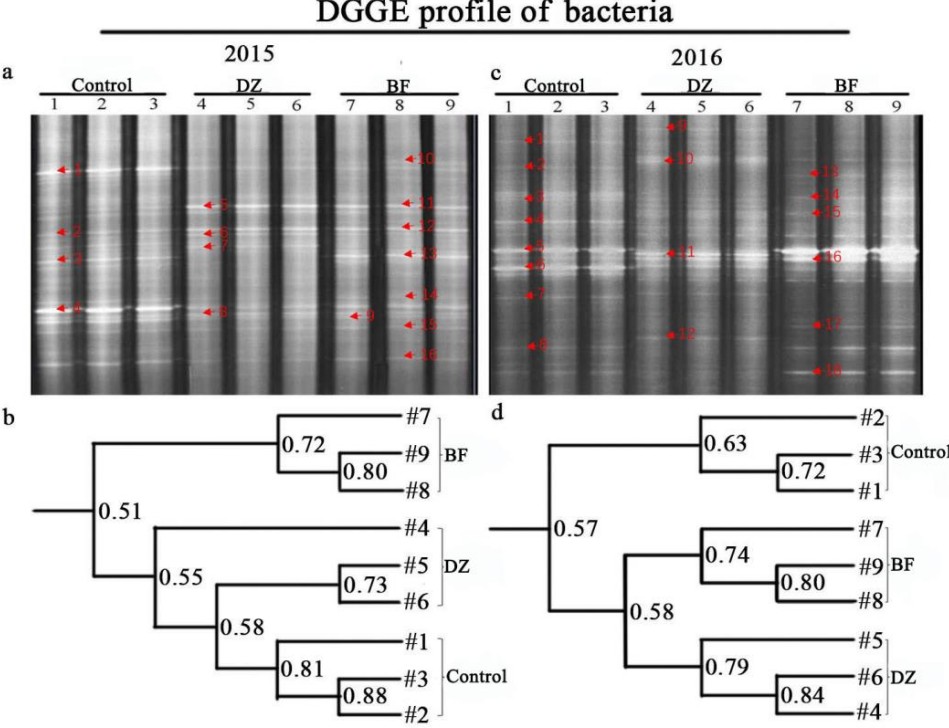

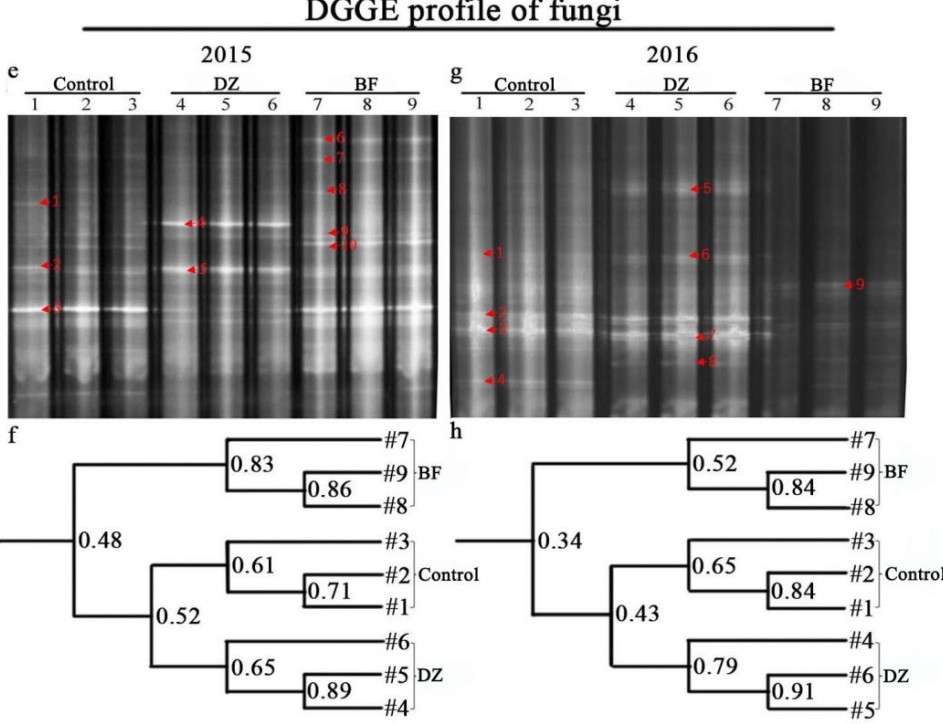

**Figure 4.** Denaturing gradient gel electrophoresis (DGGE) profiling of the soil microbiota: (**a**) and (**c**) bacterial community in 2015 and 2016, (**b**) and (**d**) UPGMA cladograms based on the Dice similarity of the composition of bacterial communities in chrysanthemum rhizosphere soil in 2015 and 2016. (**e**) and (**g**) fungal community in 2015 and 2016; (**f**) and (**h**) UPGMA cladograms based on the Dice similarity of the composition of fungal communities in chrysanthemum rhizosphere soil in 2015 and 2016, "#" indicated the number of lanes in the DGGE profile. Note: numbered fragments were excised, reamplified, and sequenced.

The number of fragments found in the DGGE profiles of bacteria ranged from 22 to 30 in 2015 and from 21 to 23 in 2016 (Table 3). In the bacterial community, the range in $H'$ was from 3.12 to 3.42 in 2015 and from 3.06 to 3.14 in 2016. This suggested a small change in the bacterial community. In the fungal community, on the other hand, $H'$ ranged from 2.81 to 2.95 in 2015 and from 2.45 to 2.91 in 2016. BF treatment had the highest bacterial $H'$ value (3.14) in 2016, while the highest fungal $H'$ value (2.95) was from the control soil in 2015. The $J$ value was higher in the bacterial community than in the fungal community. No treatments showed significant differences in $J$ related to the bacterial and fungal evenness. The S value were significantly decreased in the DZ and BF treatment when compared with the control in 2015. In 2016, the S value of the bacterial community showed no significant differences in all treatment, while the S value of the fungal community was significantly decreased in the BF treatment.

**Table 3.** Effects of different fungicides on the rhizosphere soil microbial diversity index.

| Soil microbiota | Treatment | 2015 | | | 2016 | | |
|---|---|---|---|---|---|---|---|
| | | $H'$ | $J$ | S | $H'$ | $J$ | S |
| Bacteria | Control | 3.42 ± 0.05 [a] | 0.97 ± 0.05 [a] | 30.33 ± 0.58 [a] | 3.10 ± 0.01 [ab] | 0.95 ± 0.04 [a] | 22.67 ± 0.67 [a] |
| | DZ | 3.12 ± 0.03 [b] | 0.96 ± 0.18 [a] | 22.67 ± 0.58 [b] | 3.06 ± 0.03 [b] | 0.95 ± 0.14 [a] | 21.33 ± 0.67 [a] |
| | BF | 3.12 ± 0.05 [b] | 0.96 ± 0.05 [a] | 22.67 ± 1.15 [b] | 3.14 ± 0.02 [a] | 0.94 ± 0.07 [a] | 23.33 ± 0.33 [a] |
| Fungi | Control | 2.95 ± 0.02 [a] | 0.95 ± 0.13 [a] | 18.67 ± 0.33 [a] | 2.91 ± 0.05 [a] | 0.94 ± 0.13 [a] | 17.33 ± 0.33 [a] |
| | DZ | 2.81 ± 0.02 [b] | 0.94 ± 0.06 [a] | 16.67 ± 0.33 [b] | 2.85 ± 0.04 [a] | 0.94 ± 0.06 [a] | 17.67 ± 0.67 [a] |
| | BF | 2.84 ± 0.03 [b] | 0.94 ± 0.08 [a] | 17.33 ± 0.33 [b] | 2.45 ± 0.05 [b] | 0.91 ± 0.11 [a] | 11.67 ± 0.67 [b] |

Data given in the form mean ± standard deviation. Treatment codes: Control: No treatment, DZ: 30 g dazomet per m$^2$ soil, BF: 30 g *Bacillus subtilis*-enhanced BF per m$^2$ soil. Different letters indicate significant differences among soil treatments according to Duncan's multiple range test at $p < 0.05$ level, [a], [b] and [c] indicated different levels of significant difference.

For bacterial sequence analysis, all 16 and 18 variant bands were selected, isolated, and sequenced from the DGGE gel with 2015 and 2016 samples. Four sequences in 2015 and 10 in 2016 shared 97% to 100% homology with known sequences (Table 4). The remaining sequences were 92%–96% homologous with uncultured sequences. Four of these sequences were sequences present in uncultured bacteria in 2015, and the others were sequences present in a *Rhizobium* sp., a *Halomonas* sp., an uncultured *Sphingobium* sp., a *Roseospira* sp., *a Rhizobiales bacterium*, and a *Mesorhizobium* sp. However, there were eight sequences present in uncultured bacteria in 2016, and 10 sequences were divided among an uncultured *Mitsuaria* sp., a *Halomonas* sp., a *Flexibacter* sp., a *Microbulbifer* sp., an *Agrobacterium* sp., a *Micromonospora* sp., a *Sporanaerobacter* sp., a *Rhizobium* sp., and a *Luteibacter* sp.

**Table 4.** Phylogeny of sequences present in the 16S rRNA amplicons (bacteria).

| Years | DGGE Band | Closest Relatives Micro-organisms | Treatment | | | Similarity (%) | Genebank Accession No. |
|---|---|---|---|---|---|---|---|
| | | | Control | DZ | BF | | |
| 2015 | 1 | *Uncultured bacterium* | + | + | + | 97 | JN802358.1 |
| | 2 | *Uncultured Chroococcidiopsis* sp. | + | + | + | 99 | LN878320.1 |
| | 3 | *Rhizobium* sp. | + | − | + | 98 | KP299211.1 |
| | 4 | *Halomonas* sp. | + | − | − | 95 | KX001824.1 |
| | 5 | *Roseospira* sp. | - | + | − | 96 | LT158757.1 |
| | 6 | *Uncultured sphingobium* | - | + | − | 99 | HM371404.1 |
| | 7 | *Rhizobium* sp. | - | + | − | 100 | EU689093.1 |
| | 8 | *Uncultured bacterium* | + | + | + | 97 | GQ289422.1 |
| | 9 | *Uncultured alpha proteobacterium* | - | − | + | 95 | LC017178.1 |
| | 10 | *Rhizobiales bacterium* | - | − | + | 92 | KU531579.1 |
| | 11 | *Uncultured bacterium* | - | − | + | 95 | KU928400.1 |
| | 12 | *Uncultured bacterium* | - | − | + | 97 | JX673167.1 |
| | 13 | *Uncultured rhizobiales bacterium* | - | − | + | 98 | JF731367.1 |
| | 14 | *Mesorhizobium* sp. | - | − | + | 94 | DQ659080.1 |
| | 15 | *Uncultured bacterium* | + | + | + | 98 | KT791490.1 |
| | 16 | *Uncultured Mariniflexile* sp. | + | + | + | 92 | KM108655.1 |

**Table 4.** *Cont.*

| Years | DGGE Band | Closest Relatives Micro-organisms | Treatment | | | Similarity (%) | Genebank Accession No. |
|---|---|---|---|---|---|---|---|
| | | | Control | DZ | BF | | |
| 2016 | 1 | *Uncultured Mitsuaria* | + | − | − | 97 | KM052508.1 |
| | 2 | *Halomonas* sp. | + | − | − | 97 | KX001824.1 |
| | 3 | *Uncultured flexibacter* sp. | + | − | − | 98 | EU926966.1 |
| | 4 | *Uncultured bacterium* | + | − | − | 96 | KY970042.1 |
| | 5 | *Uncultured Mariniflexible* sp. | + | − | − | 92 | KM108655.1 |
| | 6 | *Uncultured Microbulbifer* sp. | + | + | + | 99 | LN846107.1 |
| | 7 | *Uncultured Agrobacterium* sp. | + | + | - | 100 | KU375567.1 |
| | 8 | *Uncultured bacterium* | + | − | + | 98 | AY649352.1 |
| | 9 | *Micromonospora* sp. | - | + | − | 98 | KP900783.1 |
| | 10 | *Uncultured bacterium* | - | + | + | 92 | MF079697.1 |
| | 11 | *Uncultured alpha proteobacterium* | - | + | + | 97 | LC017178.1 |
| | 12 | *Uncultured actinobacterium* | - | + | − | 91 | FJ620849.1 |
| | 13 | *Uncultured Micromonospora* sp. | - | − | + | 92 | KP900783.1 |
| | 14 | *Uncultured bacterium* | - | − | + | 100 | JF261711.1 |
| | 15 | *Uncultured Sporanaerobacter* sp. | - | − | + | 100 | KR064311.1 |
| | 16 | *Uncultured Rhizobium* sp. | - | − | + | 99 | GQ365753.1 |
| | 17 | *Uncultured bacterium* | - | − | + | 93 | FJ753412.1 |
| | 18 | *Uncultured Luteibacter* sp. | + | − | + | 98 | JQ027706.1 |

Treatment codes: Control: No treatment, DZ: 30 g dazomet per m$^2$ soil, BF: 30 g *Bacillus subtilis*-enhanced BF per m$^2$ soil. "+": The micro-organism existed in the soil of the corresponding treatment. "−": The micro-organism was inexistent in the soil of the corresponding treatment.

For fungal sequence analysis, all 10 and nine variant bands were selected and sequenced from the DGGE gel of 2015 and 2016, respectively, and six in 2015 and five in 2016 shared 94% to 100% homology with known sequences (Table 5). The remaining sequences were uncultured. Four sequences were present in uncultured fungus in 2015, and the other sequences were sequences present in *F. oxysporum*, *Rhizoctonia zeae*, a *Paraphoma* sp., a *Malbranchea* sp., *Monacrosporium psychrophilum* or *Orbilia fimicola*. However, there were five sequences present in uncultured fungus in 2016, and only six dominant sequences were found originating in *Rhizoctonia zeae*, an *Anguillospora* sp., uncultured Ascomycota, *Rhizoctonia solani*, a *Tiarosporella* sp. or a *Deconica* sp.

**Table 5.** Phylogeny of sequences present in the 18S rRNA amplicons (fungi).

| Years | DGGE Band | Closest Relatives Microorganisms | Treatment | | | Similarity (%) | Genebank Accession No. |
|---|---|---|---|---|---|---|---|
| | | | Control | DZ | BF | | |
| 2015 | 1 | *Uncultured fungus* | + | − | − | 95 | HM246432.1 |
| | 2 | *Fusarium oxysporum* | + | − | − | 98 | KR063173.1 |
| | 3 | *Rhizoctonia zeae* | + | + | + | 100 | KT347101.1 |
| | 4 | *Paraphoma* sp. | − | + | − | 94 | LC126020.1 |
| | 5 | *Uncultured Orbiliaceae* | + | + | + | 96 | KF258906.1 |
| | 6 | *Uncultured eukaryote* | − | − | + | 97 | KF357450.1 |
| | 7 | *Uncultured eukaryote* | − | − | + | 97 | AB627976.2 |
| | 8 | *Malbranchea cinnamomea* | + | − | + | 96 | JX268593.1 |
| | 9 | *Monacrosporium* sp. | + | + | + | 94 | AJ001998.1 |
| | 10 | *Orbilia fimicola* | - | − | + | 92 | AF006307.1 |
| 2016 | 1 | *Anguillospora furtiva* | + | − | − | 100 | AY357262.1 |
| | 2 | *Rhizoctonia zeae* | + | + | − | 100 | KT347101.1 |
| | 3 | *Uncultured Ascomycota* | + | + | − | 99 | FJ889079.1 |
| | 4 | *Rhizoctonia solani* | + | − | − | 98 | JF499071.1 |
| | 5 | *Tiarosporella graminis* | - | + | − | 99 | KF531827.1 |
| | 6 | *Uncultured eukaryote* | - | + | − | 96 | KU657636.1 |
| | 7 | *Deconica* sp. | - | + | − | 99 | KJ137262.1 |
| | 8 | *Uncultured fungus* | - | + | + | 98 | KJ755398.1 |
| | 9 | *Uncultured eukaryote* | + | + | + | 99 | KF357450.1 |

Treatment codes: Control: No treatment, DZ: 30 g dazomet per m$^2$ soil, BF: 30 g *Bacillus subtilis*-enhanced BF per m$^2$ soil. "+": The micro-organism existed in the soil of corresponding treatment. "−": The micro-organism was inexistent in the soil of corresponding treatment.

## 4. Discussion

*Fusarium* wilt is an economically important disease for chrysanthemum greenhouse managers in China, and is detrimental to the floral industry through extensive quality loss and increased fungicide usage [1,34]. Fungicide application is the most effective current intervention against soilborne pathogens. In the current study, we confirmed that both DZ and BF can effectively reduce the incidence of *Fusarium* wilt of the chrysanthemum '*Jimba*.' Treatment with BF reduces the incidence of *Fusarium* wilt to a lesser extent than DZ in the first study year (2015). This difference may have been due to the presence of a high pathogen load in the soil, given that chrysanthemum had been continuously cropped at the site for many years; thus, any beneficial effect of the biofungicide on the growth of the pathogen antagonists may have been insufficient to overcome pathogen pressure [2].

The DZ and BF lowered the DI and heightened the DRP of chrysanthemum in two years of evaluation, but the application of DZ was significantly inhibitory to the urease and catalase enzyme activities in both 2015 and 2016. In contrast, the application of BF significantly enhanced urease and acid phosphatase activities. This effect is likely related to the activity of pathogen antagonists that thrive in the BF and the production of antibiotics by *Bacillus subtilis* [5]. Certain peptides produced by *Bacillus subtilis* have been demonstrated to exhibit antibiosis against *F. oxysporum* and other pathogenic fungi [35].

DGGE-based diversity profiling indicated that the application of DZ or BF significantly affected the structure of the soil bacterium and fungus communities. DZ and BF treatments significantly decreased the number of fungal DGGE fragments. These suggested that fungal-species richness was compromised. The same conclusions could be drawn from the behavior of other diversity metrics, $H'$ and $J$, supporting the observation reported by Qiu et al. that biocontrol-agent application encourages the growth of *Bacillus subtilis* but decreases the level of fungal diversity [36]. This may account for the colonization of *Bacillus subtilis* in the rhizosphere and the biofilm secreted by *Bacillus subtilis* with antagonistic ability decreased the fungal diversities [37]. As such, lower fungal diversity and a balanced microbial ecosystem in the amended soil play an important role in the capacity of soils to suppress soilborne plant disease [38].

The DZ treatment significantly increased the B/F ratio in the first evaluation year (2015), which reached 4.42-fold of that of the control (Table 2), while it showed a decrease in the second evaluation year (2016). Possible explanations for this behavior are that DZ is a broad-spectrum chemical fungicide, and continuous application can inhibit fungal growth, affecting the B/F ratio [39]. The BF treatment significantly increased the B/F ratio in 2015 and 2016. The result is similar to our previous research [2]. It is mainly attributed to the possibility that antibiotics were produced by antagonistic micro-organisms.

In our study, soil health refers to the high level of steady-state on soil-enzyme activities and low populations of plant pathogens. As a result, disease suppression can function as an indicator for a stable and healthy soil ecosystem, and microbial structural and functional diversity in soil may be important for soil health. The application of BF and DZ significantly changed the composition of the soil microbe. Most of the 34 sequenced bands (2015 and 2016) of 16S rDNA had sequences related to soil bacteria, such as *Rhizobium* spp., *Halomonas* spp., *Roseospira* spp., *Rhizobiales bacterium*, *Mesorhizobium* spp., *Thalassospira*, and *Sporanaerobacter* spp. A *Sphingobium* sp. and a *Rhizobium* sp. were the dominant bacteria in the DZ treatment. According to the literature, a *Sphingobium* sp. is an efficient pesticide-decomposition strain [40], which is similar to the studies that showed that the bacterial strain of Sphingobium sp. MP9-4, isolated from petroleum-contaminated soil, was efficient in degrading 1-methylphenanthrene [41]. It is the dominant bacteria in the DZ treatment probably due to the presence of chemical residues after the DZ treatment. *Rhizobium* sp. is a type of strain with nitrogen fixation [42] that has the capacity to convert atmospheric dinitrogen (N2) to a reduced form, and provide more nitrogen for plant growth. *Rhizobium* sp. is also the dominant bacteria in the DZ treatment probably due to the significantly decreased of the abundance of *F. oxysporum* in DZ treatments, then induced a reassembly of conducive community from an originally

*Fusarium*-dominated community. Similarly, *Rhizobiales bacterium* and a *Mesorhizobium* sp. were the dominant bacteria in the BF treatment. *Rhizobiales bacterium* and *Mesorhizobium* sp. were efficient biological nitrogen fixation and phosphate solubilization strain [43]. On the basis of the fungal-specific 18S rDNA assay, most of the 19 sequenced bands (2015 and 2016) correlated to *F. oxysporum*, *Rhizoctonia zeae*, *Rhizoctonia solani*, a *Deconica* sp., a *Monacrosporum* sp., *Anguillospora*, *Tiarosporella*, and a *Paraphoma* sp. Among them, *F. oxysporum*, *Rhizoctonia zeae*, and *Rhizoctonia solani* were the dominant fungi in the control rhizosphere soil, and they are plant pathogens that cause significant productivity loss.

## 5. Conclusion

In summary, the biofungicide in the control of *Fusarium* wilt is a promising and environmentally friendly strategy. In particular, the BF treatment significantly increased the abundance of *Rhizobiales* bacterium and a *Mariniflexile* sp., and decreased the abundance of *F. oxysporum* when compared with DZ after two years continuous application. These findings suggest that the overall effect of the rhizosphere soil biota on chrysanthemum was more positive in the BF treatment than in the DZ treatment. We now know more about the composition of microbe communities in rhizosphere soil after chemical fungicide and biofungicide application, which can lead to a better understanding of microbe roles in soil ecosystems.

**Author Contributions:** Conceptualization, W.F.; funding acquisition, F.C. and W.F.; investigation, H.C., J.Z., and J.J.; methodology, S.Z. and K.Z.; project administration, S.Z., F.C., and W.F.; writing—original draft, H.C.; writing—review and editing, H.C. and S.Z.

**Funding:** This research was supported by the Fund for Independent Innovation of Agricultural Sciences in Jiangsu Province (CX(18)2020), the Project for Agricultural Technology Extension Service Pilot from Scientific Research Institute in Jiangsu Province (TG(17)002), the Fundamental Research Funds for the Central Universities (KYZ201833), and the Policy Guidance Program of Jiangsu Province (BY2016077-06).

**Acknowledgments:** We thank the workers in Chinese chrysanthemum germplasm resources conservation 430 center. They provided methods of greenhouse management.

**Conflicts of Interest:** The authors declare no conflict of interest.

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
