# Peer review of "Evaluation of Soil-Applied Chemical Fungicide and Biofungicide for Control of the Fusarium Wilt of Chrysanthemum and Their Effects on Rhizosphere Soil Microbiota"

_agriculture, doi:10.3390/agriculture8120184_

Reviewer 1 Report

The effects of soil-applied chemical fungicide and biofungicide on Fusarium wilt and the soil enzyme activities and microbe of chrysanthemum were investigated. The aim of this study have been acquisition guidance for the control of Fusarium wilt of chrysanthemum, improve the knowledge of the composition of microbe  communities in the rhizosphere soil after chemical fungicide and biofungicide application and lead to a better understanding of microbe roles in these soil ecosystems.

The research was well done in terms of methodology. Correct laboratory techniques were used. The manuscript brings new values to the discipline of agriculture. It should be printed in the journal Agriculture after minor adjustments.

Comments

1. In Figures 2 and 3, homogeneous groups were given for years of research, and should be given for objects.

2. References requires ordering and uniform citation, e.g. items 25, 28 and 29 do not have a number and pages.

Author Response

Response to Reviewer 1 Comments

Point 1: In Figures 2 and 3, homogeneous groups were given for years of research, and should be given for objects.

Response 1: Thanks for your suggestion. As the reviewer 1 suggested, homogeneous groups were given for objects in Figures 2 and 3 (revised Figure 2 in line 234, Figure 3 in line 272).

Point 2: References requires ordering and uniform citation, e.g. items 25, 28 and 29 do not have a number and pages.

Response 2: Thank you for your suggestion. The references were checked all items and items 9, 12, 13, 25, 28, 29, 37, 38 and 41 were made modifications.

Reviewer 2 Report

The paper describe a two-year study on chemical and biological fungicide treatment of chrysanthemum to evaluate their protection against Fusarium wilt caused by F. oxysporum. The manuscript is reasonably well-written, however, I have some basic concerns about the design and scientific value of the study.

First of all, there is no clear scientific question or problem to solve. I feel that this is only a simple experiment to test chemical and biological agents. There is no in-depth comparison of their efficiencies and mode of action. Also rhizosphere microbiome comparison seems shallow and was not discussed in detail concerning the possible meaning or implications of the results on biology or ecology of the organisms studied. What does ‘soil health’ actually state for in the context of the study? This issue also deserves proper explanation. How the Authors can be sure that only F. oxysporum applied was responsible for disease symptoms development? I would suggest to re-write the section concerning the DGGE method for identification of the soil microbiota. Fig. 4 not clear itself, it is difficult to compare these results as the actual identification (Tables 4 and 5) comes much later. Moreover, from Table 2 one can learn that the amounts of fungi in soil are much higher for control than for all other variants, indicating that the fungal community is rich. Does it relate to the disease development? I suggest to address this issue, as it could be possible that antagonistic microorganisms are already present in the soil and additional treatment may be not necessary to protect the plants from the pathogen? Or, as the Authors suggest in Discussion, the pathogen load was so high, then it would be useful to confirm it in controls also using F. oxysporum-specific qPCR assay. In my opinion, the improvement of chrysanthemum quality upon BF treatment (section 3.3.) is interesting and deserves a closer look, particularly while there were large differences in disease incidences in both seasons and also ambiguous results were observed for both treatments (fig. 2).

Author Response

Response to Reviewer 2 Comments

Point 1: There is no clear scientific question or problem to solve. I feel that this is only a simple experiment to test chemical and biological agents. There is no in-depth comparison of their efficiencies and mode of action. 

Response 1: Chemical fungicides are frequently used interventions for control of Fusarium wilt of chrysanthemum growers in China. Now, the continuous application of chemical fungicides caused a serious soil problem, such as soil chemical residue, soil salinization, imbalance of soil microbial community structure and so on. Nowadays, bio-fungicide has been reported has advantaged environmentally friendly disease control efficacy. However, less study has been aroused to compare the effects of in used chemical and bio-fungicides on the soil microbiota of chrysanthemum. Thus, the results of our study can not only provide a valuable guidance for control the Fusarium wilt to chrysanthemum growers but also lead to a better understanding of microbe roles in these two fungicides applied soil ecosystems.

Point 2: Rhizosphere microbiome comparison seems shallow and was not discussed in detail concerning the possible meaning or implications of the results on biology or ecology of the organisms studied.

Response 2: Thanks for your suggestion. Rhizosphere microbiome comparison was rediscussed. Please see line 392-415 in the new revision.

Point 3: What does ‘soil health’ actually state for in the context of the study? This issue also deserves proper explanation.

Response 3: Based on our study, the ‘soil health’ refers to the high level of steady-state on soil enzyme activities and low populations of plant pathogens, which can maintain sustainable chrysanthemum industrial development.

Point 4: How the Authors can be sure that only F. oxysporum applied was responsible for disease symptoms development?

Response 4: In our previous works, we have done a lot of isolation an identification works, we found that F. oxysporum and F. solani were the major pathogenic species of chrysanthemum (Song et al., 2013). After that, we did Koch's test (re-inoculate and re-isolation), and the result confirmed that Fusarium oxysporum f. sp. chrysanthemi is the main agent of chrysanthemum Fusarium wilt (Zhao et al., 2016).

Point 5: I would suggest to re-write the section concerning the DGGE method for identification of the soil microbiota.

Response 5: We have re-write the section concerning the DGGE method for identification of the soil microbiota (revised manuscript, line 155-173).

Point 6: Fig. 4 not clear itself, it is difficult to compare these results as the actual identification (Tables 4 and 5) comes much later.

Response 6: We have reorganized the Table 4 and Table5 in the revised manuscript according to your comment (Table 4 in line 339, Table 5 in line 351).

Point 7: From Table 2 one can learn that the amounts of fungi in soil are much higher for control than for all other variants, indicating that the fungal community is rich. Does it relate to the disease development? I suggest to address this issue, as it could be possible that antagonistic microorganisms are already present in the soil and additional treatment may be not necessary to protect the plants from the pathogen? Or, as the Authors suggest in Discussion, the pathogen load was so high, then it would be useful to confirm it in controls also using F. oxysporum-specific qPCR assay.

Response 7: Disease development is close related to the abundance of fungi, and the ratio of bacterial/fungi is determinative for soil health. In our study, the abundance of F. oxysporum was significantly decreased in DZ and BF treatments, probably then induced a reassembly of antagonistic conducive community from an originally Fusarium-dominated community, which may have played an important role in disease control. And the antagonistic microorganisms may be already present in the control soil, but it didn’t indicate the antagonistic microorganisms can protect the plants from the pathogen, only the abundance of antagonistic microorganisms reached a level, the antagonistic microorganisms can play the antagonistic role.

Point 8: The improvement of chrysanthemum quality upon BF treatment (section 3.3.) is interesting and deserves a closer look, particularly while there were large differences in disease incidences in both seasons and also ambiguous results were observed for both treatments (Fig. 2).

Response 8: The mechanism of improvement in chrysanthemum quality upon BF treatment in our study probably due to the application of bio-fungicide could (1) preserve the balance of soil microbiome, ensured the soil healthy; (2) increase the abundance of beneficial microorganism, such as nitrogen fixation strain-Rhizobiales bacterium, it can provide more nitrogen for chrysanthemum growth; (3) decrease the abundance of plant pathogen, such as Fusarium oxysporum, reduced the incidence of chrysanthemum Fusarium wilt.

Round  2

Reviewer 2 Report

The revised version of the manuscript reads much better. I found the Authors' explanations satisfactory and the manuscript can be processed further.